# The Effects of Antiperspirant Aluminum Chlorohydrate on the Development of Antibiotic Resistance in *Staphylococcus epidermidis*

**DOI:** 10.3390/microorganisms11040948

**Published:** 2023-04-05

**Authors:** Ayse Aras, Suna Sibel Rizvanoglu, Elif Seren Tanriverdi, Basar Karaca, Mujde Eryilmaz

**Affiliations:** 1Turkish Medicines and Medical Devices Agency, Cosmetic Products Department, Ankara 06500, Türkiye; 2Department of Pharmaceutical Microbiology, Faculty of Pharmacy, Ankara University, Ankara 06100, Türkiye; 3Department of Medical Microbiology, Faculty of Medicine, Inonu University, Malatya 44210, Türkiye; 4Department of Biology, Faculty of Science, Ankara University, Ankara 06100, Türkiye

**Keywords:** aluminum chlorohydrate, antibiotic resistance, minimum inhibitory concentration, quantitative reverse transcriptase PCR, *Staphylococcus epidermidis*

## Abstract

This study investigates the effects of the antiperspirant aluminum chlorohydrate on the development of antibiotic resistance in commensal *Staphylococcus epidermidis* isolates. The isolates were exposed to aluminum chlorohydrate for 30 days. The bacteria that developed resistance to oxacillin and ciprofloxacin were isolated, and the expression levels of some antibiotic resistance genes were determined using quantitative reverse transcriptase PCR. Before and after exposure, the minimum inhibitory concentration (MIC) values of the bacteria were determined using the microdilution method. A time-dependent increase was observed in the number of bacteria that developed resistance and increased MIC values. Consistent with the ciprofloxacin resistance observed after exposure, an increase in *norA*, *norB/C*, *gyrA*, *gyrB*, *parC*, and *parE* gene expression was observed. In addition to aluminum chlorohydrate exposure, oxacillin resistance was observed in all test bacteria in the group only subcultured in the medium, suggesting that phenotypic resistance cannot be correlated with chemical exposure in light of these data. The increase in *mecA* gene expression in selected test bacteria that acquired resistance to oxacillin after exposure compared with control groups suggests that the observed resistance may have been related to aluminum chlorohydrate exposure. To our knowledge, this is the first time in the literature that the effects of aluminum chlorohydrate as an antiperspirant on the development of antibiotic resistance in *Staphylococcus epidermidis* have been reported.

## 1. Introduction

Antibiotic resistance is one of the most serious global public health problems today. If the necessary precautions are not taken, ten million people are expected to die annually from antibiotic-resistant infections by 2050. In addition to the inappropriate use of antibiotics in humans and animals for therapeutic purposes, the unnecessary and improper use of antibiotics, especially in the food and agricultural sectors for economic reasons, is one of the main reasons for the increase in antibiotic resistance. For this reason, precautions have been taken in antibiotic consumption in the fight against antibiotic resistance. Considering the increasing resistance rates, it is seen that the measures taken are not sufficient to solve the problem. New perspectives with more holistic approaches are needed to solve this problem [1,2,3,4,5,6].

Our body is rapidly colonized with microorganisms (bacteria, fungi, viruses, etc.) from the moment of birth. The community of these microorganisms is referred to as the human microbiome. The microbiota provides many benefits to the host [7,8]. *Staphylococcus epidermidis*, a member of the normal commensal skin flora of the human body, is primarily found in the axilla, on the head, and in the nostril. *S. epidermidis* has a significant role in the skin ecosystem. It protects and prevents microbiota imbalance by fighting pathogens and participating in skin homeostasis through the production of beneficial bacterial metabolites. The widespread colonization of this bacterium on human skin and mucous membranes provides the opportunity for this bacterium to become infectious if suitable conditions are created [9,10,11]. The pathogenicity of *S. epidermidis* is mainly due to its ability to form biofilms on medical devices such as indwelling catheters and implanted devices. *S. epidermidis* can cause infections of prosthetic joints, vascular grafts, central nervous system shunts, surgical sites, and cardiac devices. The eradication of *S. epidermidis* infections is difficult because the bacteria in the biofilm are protected from immune system attack and antibacterial treatment. In recent years, *S. epidermidis* has become the most important cause of nosocomial infections [9,12,13]. It has been reported that resistance against many antibiotics, such as methicillin, erythromycin, ciprofloxacin, and gentamicin, develops in *S. epidermidis* strains isolated as nosocomial infection agents today [14,15,16,17].

Cosmetics, used by individuals from almost all age groups, have a wide range of products. There are various chemicals in the content of these products. Repeated and long-term exposure to cosmetics, many of which are used daily, can cause various effects on the organism and the microbiome [18,19,20]. Antiperspirants are cosmetic products widely used in society to prevent sweating. Many antiperspirants (aerosol, roll-on, stick, and cream), generally used daily, contain aluminum salts for this purpose. Aluminum chlorohydrate is an inorganic salt consisting of complex basic aluminum chloride. Its pH is about 4.5. This compound does not cause skin irritation, because it is less acidic than other aluminum salts. Studies have reported that exposure to aluminum compounds from cosmetics is associated with chest diseases, breast cancer, and Alzheimer’s disease [18,20,21,22]. According to the literature review, no study was found investigating the effects of aluminum chlorohydrate on the skin microbiota.

Recent studies show that some non-antibiotic long-term medications may play a role in the development of antibiotic resistance [2,6]. In addition to non-antibiotic drugs, the possibility that cosmetics used daily may play a role in the development of antibiotic resistance should also be considered. The purpose of this study is to investigate the effects of aluminum chlorohydrate, commonly used in deodorants for its antiperspirant properties, on the development of antibiotic resistance in *S. epidermidis*, a member of the normal human axillary flora. According to the literature review, this study is the first to demonstrate the effect of aluminum chlorohydrate on the development of antibiotic resistance in *S. epidermidis*, a member of the skin microbiome.

## 2. Materials and Methods

### 2.1. Bacterial Strains and Growth Conditions

Twenty-two (oxacillin and ciprofloxacin) susceptible strains and one resistant *S. epidermidis* strain isolated from the axilla were used. *S. epidermidis* ATCC 12228 (susceptible strain) was used as the standard strain. The antibiotic susceptibility of all test bacteria were confirmed with the disc diffusion test. In the susceptibility tests, *Staphylococcus aureus* ATCC 25923 was used as a control [23].

Trypticase soy broth (Merck, Darmstadt, Germany) and trypticase soy agar (Merck, Darmstadt, Germany) were used as initial growth media. Mueller–Hinton agar (MHA) (Merck, Germany) and Mueller–Hinton broth (MHB) (Merck, Darmstadt, Germany) were used in the antibiotic susceptibility tests. Aluminum chlorohydrate (Sigma-Aldrich, Roedarmark, Germany) exposure was performed in lysogeny broth (LB) (Merck, Darmstadt, Germany). MHA plates containing 0.5 μg mL^−1^ oxacillin (Sigma-Aldrich, Roedarmark, Germany) and 4 μg mL^−1^ ciprofloxacin (Sigma-Aldrich, Roedarmark, Germany) were used to isolate antibiotic-resistant test bacteria. All test bacteria were incubated at 37 °C for 18–24 h.

### 2.2. Detection of Antibiotic Resistance Genes

The presence of the *mecA* gene (encoding PBP2a synthesis), the *gyrA* and *gyrB* genes (encoding the DNA gyrase enzyme), the *parC* and *parE* genes (encoding topoisomerase IV), and the *norA* and *norB/C* genes, which play a role in regulating the efflux pump in the test bacteria, was detected using polymerase chain reaction (PCR) [24,25,26,27,28,29]. The primers used for PCR were given in Appendix A. Gel electrophoresis was used to detect amplification products, which were visualized using SafeView Classic (ABM, Canada). The size of the PCR products was compared with a 100 bp DNA ladder (New England BioLabs, Ipswich, MA, USA).

### 2.3. Determination of Minimum Inhibitory Concentration Values

Before and after exposure to aluminum chlorohydrate, the MIC values of test bacteria against oxacillin and ciprofloxacin were determined using the microdilution method [23]. An increase in the MIC values was detected after exposure. Each test bacterium was tested in triplicate.

### 2.4. Aluminum Chlorohydrate Exposure and Isolation of Resistant Bacteria against Test Antibiotics

Considering the Scientific Committee on Consumer Safety 2018 guidelines, the daily exposure amount derived from applying non-spray antiperspirants to the armpits twice a day was set at 1.5 g. The average highest concentration of aluminum chlorohydrate in antiperspirants on the market was assumed to be 20% [30,31]. Based on this information, the daily exposure to aluminum chlorohydrate was determined to be a minimum of 300 mg/L, and the aluminum chlorohydrate exposure concentrations of bacteria were adjusted according to this value. Due to the rapid moisture absorption of aluminum chlorohydrate, aluminum chlorohydrate solution was freshly prepared with sterile distilled water in each step of the study.

First, the exposure concentration of aluminum chlorohydrate was determined using the MIC test [23]. A volume of 40 μL of suspension of the test bacteria (adjusted to 10^8^–10^9^ cfu mL^−1^) was added to 3.96 mL of LB containing aluminum chlorohydrate (300 mg/L). After incubation at 37 °C for 18–24 h, 40 μL of the test bacteria cultures was transferred to 3.96 mL of fresh LB containing the above-stated concentration of aluminum chlorohydrate and incubated again. This subculturing procedure was repeated continuously for 30 days. On the 10th and 30th days of this procedure, 100 μL of the exposed test bacteria samples was plated on the MHA plates with 0.5 μg mL^−1^ oxacillin and 4 μg mL^−1^ ciprofloxacin. After 24 h of incubation at 37 °C, colonies formed on the MHA plates containing test antibiotics were found to be resistant. Test bacteria subcultured in LB broth without aluminum chlorohydrate was used as the control group [2,3,4,5,6].

### 2.5. Quantitative Reverse Transcriptase PCR Analysis of Target Genes

The expression levels of the target genes of test bacteria were determined using quantitative reverse transcriptase PCR (RT-qPCR). The expression levels of the *mecA*, *gyrA*, *gyrB*, *parC*, *parE*, *norA*, *norB*, and *norC* genes were compared with those in the control groups [6,26,27,32]. Total RNA was extracted using an RNA isolation kit (FavorPrepTM, Ping Tung, Taiwan) according to the manufacturer’s instructions. Total RNA was quantified in each sample using a NanoDrop spectrophotometer (Thermo Scientific, Waltham, MA, USA). cDNA was synthesized using an iScriptTM cDNA reverse transcription kit (Bio-Rad, Hercules, CA, USA) according to the manufacturer’s instructions. iTaqTM Universal SYBR^®^ Green Supermix Kit (2×) (Bio-Rad^®^, Hercules, CA, USA) was used to determine the expression levels of the target genes. RT-qPCR analysis was performed on each 10 µL of PCR reaction mixture (5 µL of iTaqTM Universal SYBR^®^ Green Supermix 2×, 3 µL of ddH2O, 1 µL of cDNA template, 0.5 µL of forward primer, and 0.5 µL of reverse primer). Relative gene expression values were calculated with the 2^−ΔΔCT^ method using the RT2 Profiler PCR Array Data Analysis v3.5 (Qiagen, Hilden, Germany) analysis program. All data were normalized to 16S rRNA housekeeping gene expression levels.

## 3. Results and Discussion

### 3.1. Detection of Antibiotic Resistance Genes

Agarose gel electrophoresis images of the antibiotic resistance genes are given in Figure 1. All test bacteria were found to have *gyrA, gyrB*, *parC*, *parE, norA*, and *norB/C* genes. However, the *mecA* gene was only detected in test bacteria **1**, **12**, **13**, **14**, **15**, **16**, **22**, and **23**.

### 3.2. MIC Values of Aluminum Chlorohydrate

No antibacterial effect of aluminum chlorohydrate was observed against test bacteria in the range of 187.5–24,000 mg L^−1^. Therefore, the previously reported concentration of 300 mg L^−1^ determined as the exposure concentration of aluminum chlorohydrate for the test bacteria was considered appropriate.

### 3.3. Antibiotic Resistance Development after Aluminum Chlorohydrate Exposure

The number of susceptible test bacteria that developed phenotypic resistance to oxacillin and ciprofloxacin on day 10 and day 30 of exposure is shown in Figure 2. The resistant test bacteria are listed in Table 1.

When considering the results of phenotypic resistance development after exposure, it was found that the number of test bacteria that developed resistance to ciprofloxacin increased. Strain **24** (*S. epidermidis* ATCC 12228), which developed resistance to ciprofloxacin after 30 days of exposure, also developed resistance in the control group, which was only subcultured in the medium for 30 days. When considering the relative expression results between the two test conditions, increased gene expression of *norA*, *norB/C*, *gyrA*, *gyrB*, *parC*, and *parE* was observed in *S. epidermidis* ATCC 12228 exposed to aluminum chlorohydrate, which is consistent with phenotypic resistance. This result suggests that the observed ciprofloxacin resistance was developed as a consequence of exposure to aluminum chlorohydrate (Figure 3, Table 2).

The development of ciprofloxacin resistance in isolate **14**, which was only subcultured in LB broth without aluminum chlorohydrate for 30 days and did not develop resistance to ciprofloxacin as a result of exposure, suggests that long-term subculturing also played a role in the development of resistance. The relative expression results of the control group test bacteria, which developed resistance to ciprofloxacin following subculturing only in LB broth without any aluminum chlorohydrate for 30 days, and the control group test bacteria taken on the first day as the starter, support this phenotypic resistance result. Under the stated conditions, an increase in *norA*, *norB/C*, *gyrA*, *gyrB*, *parC* and *parE* gene expression was observed in test bacteria **14** and **24**, which is consistent with the development of phenotypic resistance to ciprofloxacin (Figure 4 and Table 3).

Ciprofloxacin is a bactericidal antibiotic of the fluoroquinolone class used to treat infections such as urinary tract infections and pneumonia. The mechanism of action of ciprofloxacin is the inhibition of DNA replication with the inhibition of bacterial DNA topoisomerase and DNA gyrase enzymes. The development of resistance to this group of antibiotics occurs through point mutations in genes encoding DNA gyrase and topoisomerase IV or mutations in genes regulating efflux pump proteins [33,34]. The amino-terminal domains of *GyrA* or *ParC*, covalently bound to DNA in an enzyme intermediate, are the sites where these resistance mutations are most frequently detected. Quinolone resistance has also been associated with alterations in specific domains of *GyrB* and *ParE*; however, these alterations are much less common in resistant clinical bacterial isolates than mutations in *GyrA* or *ParC* [35]. In this study, an increase in *norA*, *norB/C*, *gyrA*, *gyrB*, *parC*, and *parE* gene expression was observed, in agreement with phenotypic ciprofloxacin resistance. Although efflux pumps exhibit substrate specificity, most of them can also pump different chemical compounds out of the cell. This situation is effective in the development of multidrug resistance in bacteria. More than ten efflux pump genes have been detected in methicillin-resistant *Staphylococcus aureus* (MRSA) strains. The most frequently detected *norA* and *norB/C* genes are chromosomally encoded efflux genes that are also present in *S. epidermidis* strains. Demarco et al. (2007) reported that nearly half (54/114 strains) of the *S. aureus* isolates tested effluxed at least two structurally distinct substrates [36,37,38]. In our study, an increase in the expression levels of the *norA* and *norB/C* efflux genes was detected. This could play a role in the development of resistance to other antibiotics and lead to the emergence of multidrug-resistant strains.

Methicillin resistance occurs in two different phenotypes: homogeneous and heterogeneous. Homogeneous resistance refers to a cell population in which all cells are resistant to high concentrations of methicillin. In heterogeneous resistance, although all cells carry the *mecA* gene in the bacterial community, only a small minority of bacteria exhibit high levels of methicillin resistance [39]. The *mecA* gene is not expressed in heterogeneously resistant staphylococcal strains, and bacteria that should be resistant may be found to be susceptible in routine susceptibility testing [40]. However, isolates that do not carry the *mecA* gene have also been reported to be resistant to oxacillin [41]. In our study, oxacillin resistance was observed after 30 days in both groups only subcultured in medium and exposed to aluminum chlorohydrate. PCR revealed that the *mecA* gene was only found in some of the test bacteria (Figure 1). Since all test bacteria developed resistance to oxacillin after 30 days of exposure, changes in gene expression were determined using sampling in two of them. Test bacteria **14** and **22** were selected for this purpose. The relative expression results of the *mecA* gene of test bacteria **14** and **22**, which had developed resistance to oxacillin after 30 days of exposure to aluminum chlorohydrate, and the control group test bacteria taken on the first day are shown in Figure 5. Under the stated conditions, an increase in *mecA* gene expression was observed in test bacteria **14** and **22**, which is consistent with the development of phenotypic resistance to oxacillin (Figure 5 and Table 4).

The relative expression results of the *mecA* gene in test bacteria **14** and **22**, which developed resistance to oxacillin after 30 days of exposure to aluminum chlorohydrate, and the control group of test bacteria, which developed resistance to oxacillin after 30 days of subculturing only in LB broth without aluminum chlorohydrate, are shown in Figure 6. Under the stated conditions, an increase in *mecA* gene expression was observed in test bacteria **14** and **22**, which is consistent with the development of phenotypic resistance to oxacillin (Figure 6 and Table 5). A four-fold or greater increase in gene expression was considered significant.

Since oxacillin resistance was detected in all isolates of the control group subcultured for 30 days only in LB broth without aluminum chlorohydrate, it could not be correlated with the resistance developed after aluminum chlorohydrate exposure. However, the increase in *mecA* gene expression in the selected test bacteria that became resistant to oxacillin after exposure compared with the control groups suggests that the observed oxacillin resistance is related to aluminum chlorohydrate exposure.

### 3.4. Increase in the Minimum Inhibitory Concentration Values of the Test Bacteria

The MIC values of the bacteria before and after exposure to the test antibiotics are shown in Table 6. The LB broth without aluminum chlorohydrate was used as a control.

Considering the MIC values of all test bacteria before and after exposure, it was found that the MIC values of isolates that developed ciprofloxacin resistance as a result of exposure to aluminum chlorohydrate increased 4–32-fold. Although all test bacteria developed resistance to oxacillin as a result of exposure, it was found that the MIC values of most of them increased 4–256-fold.

Microorganisms possess remarkable adaptability to cope with changes in their ecological surroundings. In order to sustain their survival and propagation under these new conditions, they employ a variety of resistance mechanisms [42]. Although the improper use of antibiotics is considered the main cause of the development of antibiotic resistance, studies in recent years have shown that some non-antibiotic drugs may also play a role in promoting antibiotic resistance. These drugs include antidepressants. Jin et al. (2018) were the first to report that fluoxetine use induced antibiotic resistance in *Escherichia coli.* In this study, the expression of the multidrug efflux pump genes *acrB* and *acrD* increased as a result of a 10 h exposure of *E. coli* to fluoxetine, while the expression levels of outer membrane porin genes *ompF* and *ompW* decreased. The bacterium protected itself from exposure to fluoxetine by expelling the drug from the cell and preventing the drug from entering the cell. It was found that a 30-day exposure to fluoxetine caused an increase in chloramphenicol, amoxicillin, and tetracycline resistance. In addition, these mutants exhibited multiple resistance to fluoroquinolones, aminoglycosides, and beta-lactams. Another study by our research group examined the effects of fluoxetine, sertraline, and amitriptyline, commonly used antidepressants, on the development of antibiotic resistance in clinical *Acinetobacter baumannii* isolates. The isolates were exposed to fluoxetine, sertraline, and amitriptyline, respectively, for 30 days. The bacteria that developed resistance to gentamicin, imipenem, colistin, and ciprofloxacin were isolated, and the expression levels of some antibiotic resistance genes were determined using quantitative reverse transcriptase PCR. Before and after exposure, the MICs of bacteria were determined using the microdilution method. A time-dependent increase in the number of bacteria that developed resistance and increased MICs was observed. After exposure to fluoxetine and sertraline, decreases in efflux and outer membrane porin genes were observed in isolates that developed colistin resistance, whereas increases were observed in isolates that developed ciprofloxacin resistance. These observations suggest that these antidepressants have similar effects on the development of resistance. While the exposure to fluoxetine did not result in the development of resistance to imipenem, it was observed after exposure to sertraline and amitriptyline, and a common decrease in *ompA* gene expression was observed in these isolates [6]. Wang et al. (2021) [43] reported that nonsteroidal anti-inflammatory drugs (ibuprofen, naproxen and diclofenac), a lipid-lowering drug (gemfibrozil), and a β-blocker (propranolol) accelerated the spread of antibiotic resistance through plasmid-borne bacterial conjugation. In addition to non-antibiotic drugs, the possibility that daily cosmetics may play a role in developing antibiotic resistance should also be considered. Many cosmetic products contain preservatives and other chemicals designed to prevent microbial growth and prolong the product’s shelf life. Therefore, it is important to consider the potential role of cosmetic products in developing antibiotic resistance and further investigate their impact on microbial communities.

## 4. Conclusions

The response of microorganisms to chemical changes in their environment has long been one of the most important topics of research. However, concrete data on how the environmental changes that cause stress conditions for microorganisms may contribute to antimicrobial resistance are not yet available. According to the literature review, this study is the first demonstrate the effect of exposure to aluminum chlorohydrate on the development of antibiotic resistance in *S. epidermidis*, a member of the skin microbiome. Because of the repeated and long-term use of cosmetics, the effect of chemicals on the resistance mechanisms of bacteria in the microbiota should be investigated with more comprehensive phenotypic and genotypic analyses.

## Figures and Tables

**Figure 1 microorganisms-11-00948-f001:**
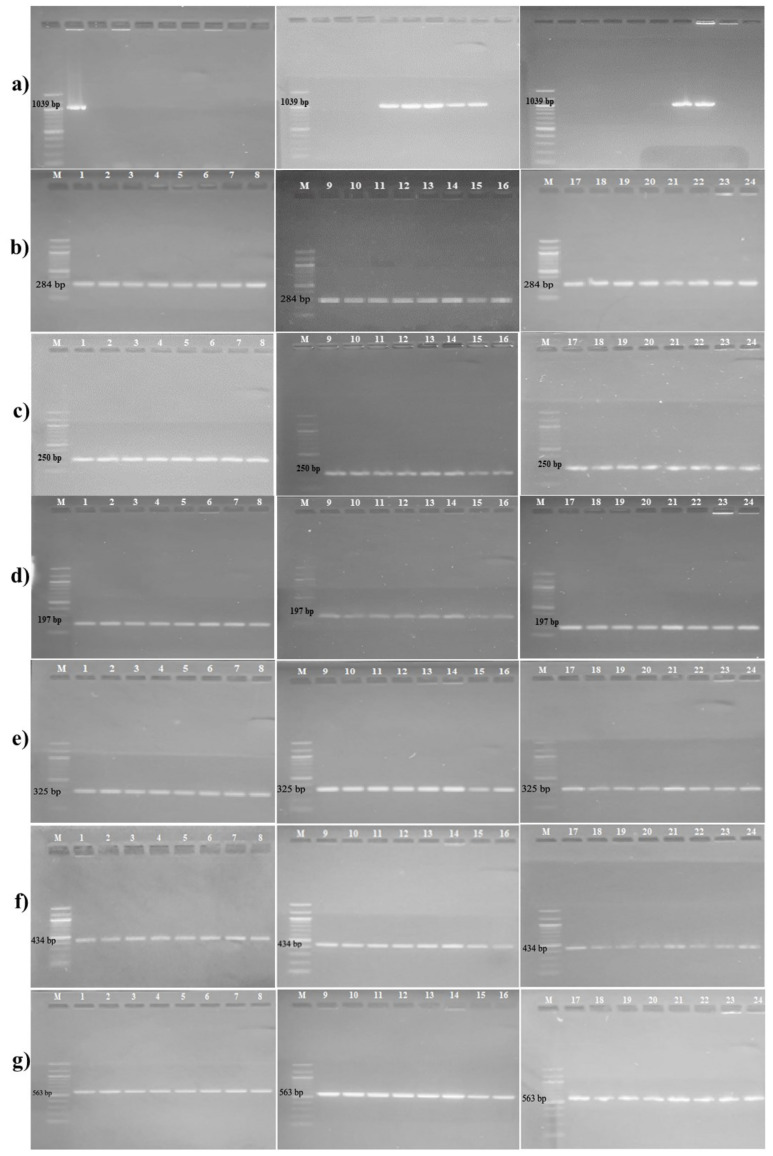
Agarose gel electrophoresis images of *mecA*, *gyrA*, *gyrB*, *parC*, *parE*, *norA,* and *norB/C* genes (M: DNA Ladder (100 bp); **1**–**22**: isolates susceptible to oxacillin and ciprofloxacin; **23**: isolate resistant to oxacillin and ciprofloxacin; **24**: *S. epidermidis* ATCC 12228). (**a**) *mecA* (1039 bp), (**b**) *gyrA* (284 bp), (**c**) *gyrB* (250 bp), (**d**) *parC* (197 bp), (**e**) *parE* (325 bp), (**f**) *norA* (434 bp), (**g**) *norB/C* (563 bp).

**Figure 2 microorganisms-11-00948-f002:**
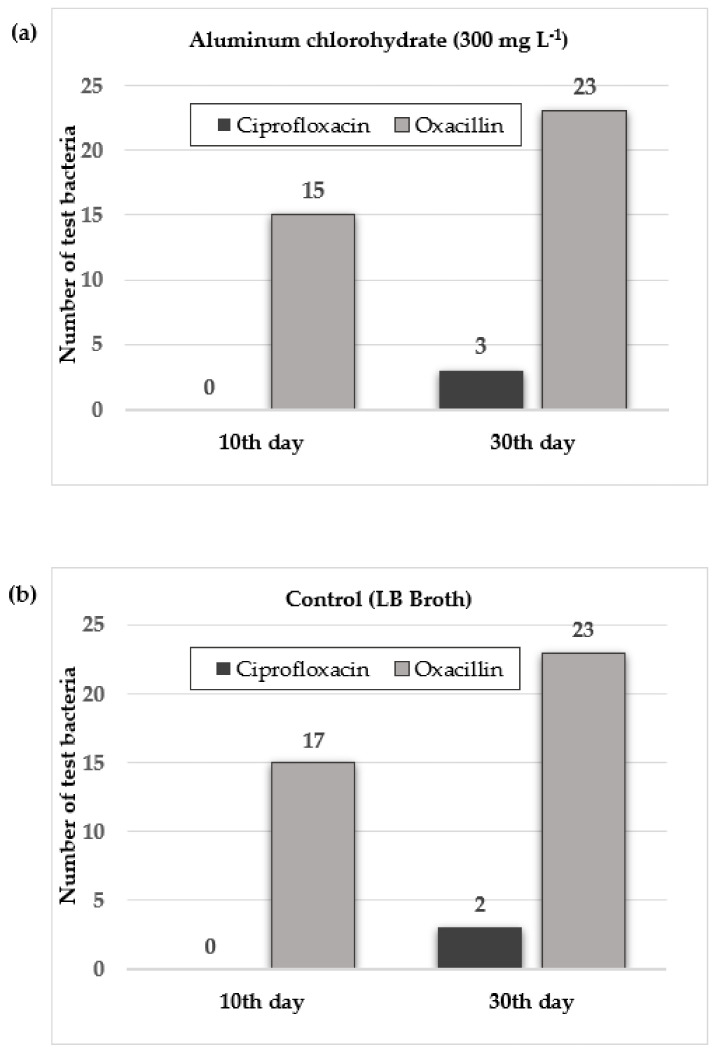
Phenotypic antibiotic resistance development of susceptible test bacteria on the 10th and 30th days of the exposure. (**a**) Aluminum chlorohydrate (300 mg/L) exposure. (**b**) Control (susceptible test bacteria subcultured in LB broth without aluminum chlorohydrate).

**Figure 3 microorganisms-11-00948-f003:**
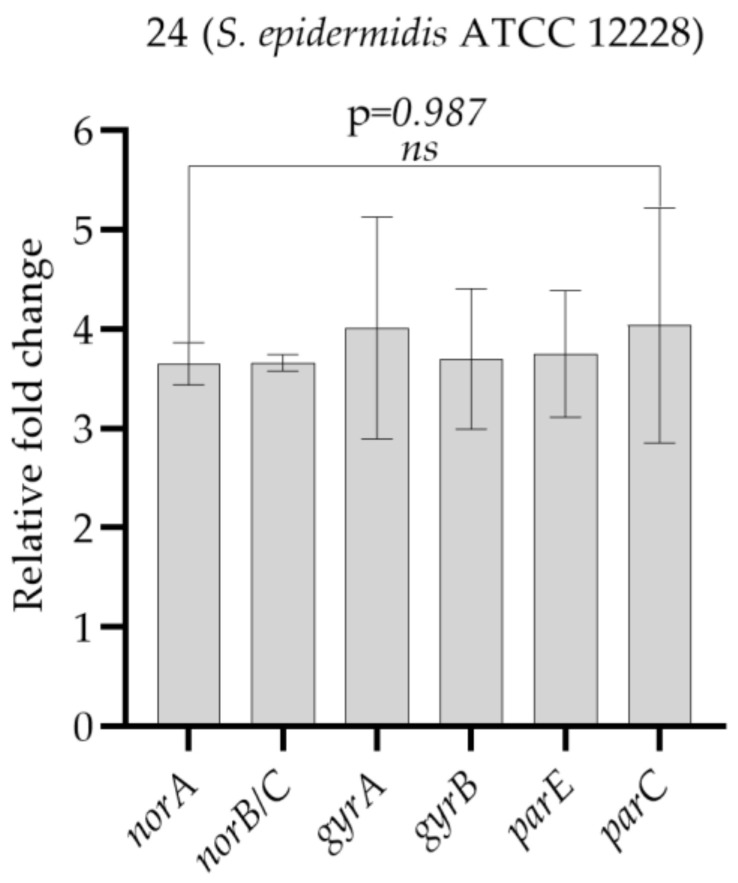
Comparison of gene expression values of strain **24** (*S. epidermidis* ATCC 12228), which developed resistance to ciprofloxacin after 30 days of exposure to aluminum chlorohydrate, and control bacterium that developed resistance to ciprofloxacin following subculturing only in LB broth without aluminum chlorohydrate for 30 days. *ns*: not significant. One-way ANOVA was used to compare the means of the groups. Tukey’s post hoc test was performed to make pairwise comparisons. Bars represent standard deviations.

**Figure 4 microorganisms-11-00948-f004:**
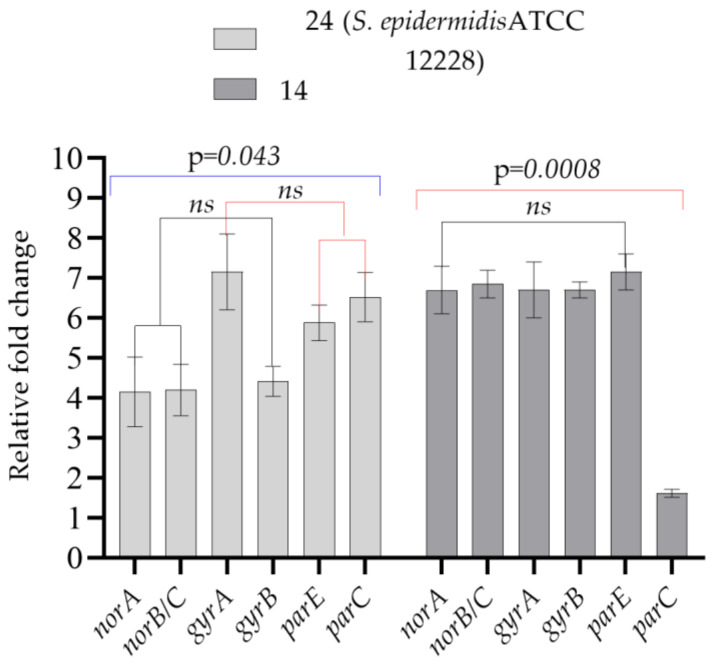
Comparison of gene expression values of the test bacteria of the control group, which developed resistance to ciprofloxacin following subculturing only in LB broth without aluminum chlorohydrate for 30 days, and the test bacteria of the control group, which were taken as starter on the first day. *ns*: not significant. One-way ANOVA was used to compare the means of the groups. Tukey’s post hoc test was performed to make pairwise comparisons. Bars represent standard deviations.

**Figure 5 microorganisms-11-00948-f005:**
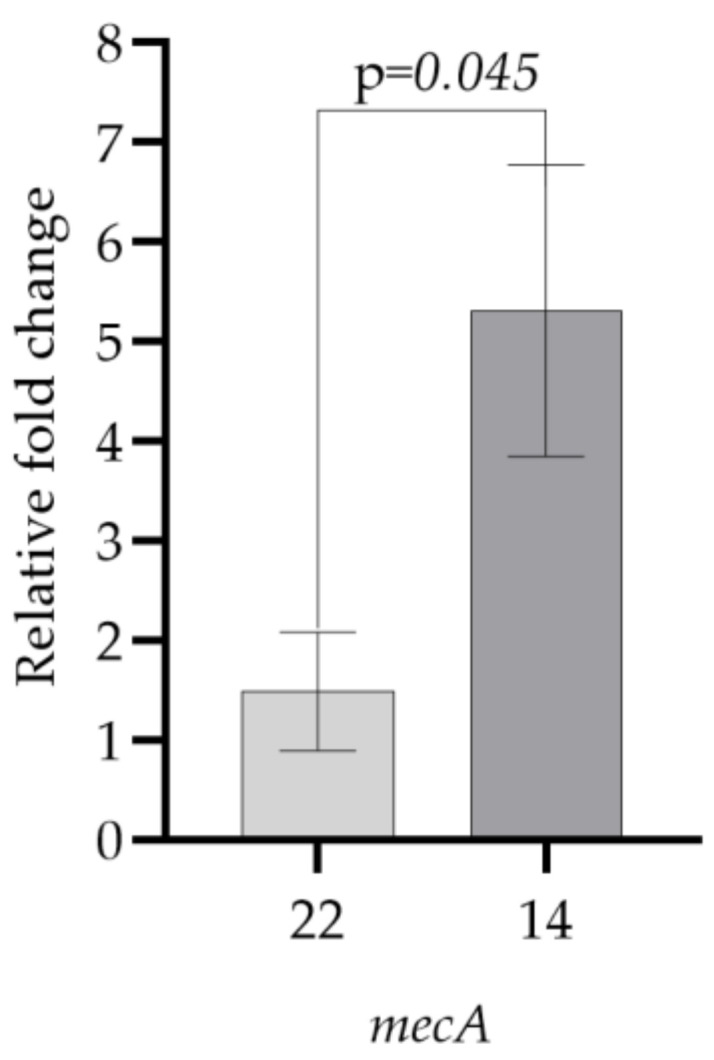
Comparison of *mecA* gene expression levels of test bacteria **14** and **22**, which developed resistance to oxacillin after 30 days of exposure to aluminum chlorohydrate, and the control group test bacteria taken on the first day as the starter (paired *t*-test).

**Figure 6 microorganisms-11-00948-f006:**
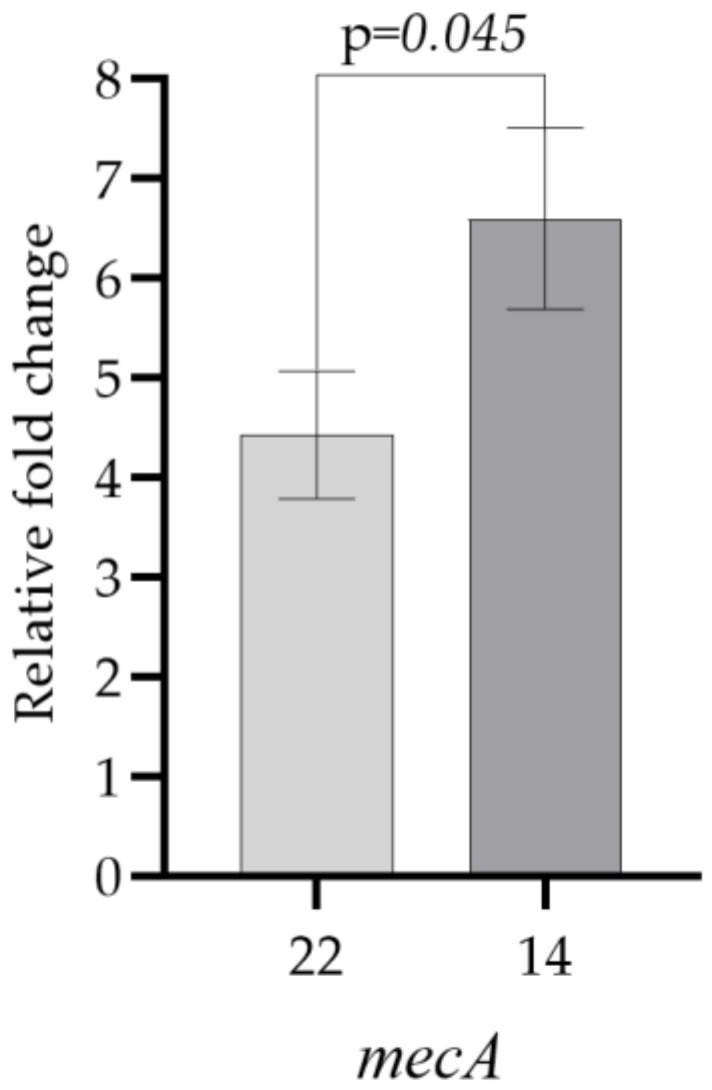
Comparison of *mecA* gene expression values of test bacteria **14** and **22**, which developed resistance to oxacillin after 30 days of exposure to aluminum chlorohydrate, and the control group test bacteria, which developed resistance to oxacillin following subculturing only in LB broth without aluminum chlorohydrate for 30 days (paired *t*-test).

**Table 1 microorganisms-11-00948-t001:** Susceptible test bacteria that developed phenotypic resistance according to the exposure times.

Exposure Day	Test Antibiotic	Aluminum Chlorohydrate (300 mg L^−1^)	Control *
10th	CIP	-	-
	OXA	**1**, **2**, **3**, **5**, **12**, **13**, **14**, **15**, **16**, **17**, **18**, **20**, **21**, **22**, **24**	**1**, **2**, **3**, **4**, **6**, **10**, **11**, **12**, **13**, **14**, **15**, **16**, **17**, **18**, **19**, **20**, **22**
30th	CIP	**18**, **22**, **24**	**14**, **24**
	OXA	**1**, **2**, **3**, **4**, **5**, **6**, **7**, **8**, **9**, **10**, **11**, **12**, **13**, **14**, **15**, **16**, **17**, **18**, **19**, **20**, **21**, **22**, **24**	**1**, **2**, **3**, **4**, **5**, **6**, **7**, **8**, **9**, **10**, **11**, **12**, **13**, **14**, **15**, **16**, **17**, **18**, **19**, **20**, **21**, **22**, **24**

**1**–**22**: Susceptible isolates; **24**: *S. epidermidis* ATCC 12228. CIP: ciprofloxacin; OXA: oxacillin. * Susceptible test bacteria that subcultured in LB broth without aluminum chlorohydrate for 30 days.

**Table 2 microorganisms-11-00948-t002:** Relative fold change in resistance gene expression levels of strain **24** (*S. epidermidis* ATCC 12228), which developed resistance to ciprofloxacin after 30 days of exposure to aluminum chlorohydrate, and control bacteria, which developed resistance to ciprofloxacin only following 30-day subculturing in LB broth without aluminum chlorohydrate.

Test Bacteria	*norA*	*norB/C*	*gyrA*	*gyrB*	*parE*	*parC*
**24** (*S. epidermidis* ATCC 12228)	3.65 ± 0.21	3.66 ± 0.08	4.01 ± 1.12	3.70 ± 0.71	3.75 ± 0.65	4.03 ± 0.95

**Table 3 microorganisms-11-00948-t003:** Relative fold change in resistance gene expression levels in the control group test bacteria, which developed resistance to ciprofloxacin following subculturing only in LB broth without aluminum chlorohydrate for 30 days, and the control group test bacteria taken on the first day as the starter.

Test Bacteria	*norA*	*norB/C*	*gyrA*	*gyrB*	*parE*	*parC*
**24** (*S. epidermidis* ATCC 12228)	4.14 ± 1.23	4.20 ± 0.91	7.15 ± 1.34	4.42 ± 0.53	5.88 ± 0.63	6.52 ± 0.88
**14**	6.70 ± 0.85	6.85 ± 0.49	6.70 ± 0.99	6.70 ± 0.28	7.15 ± 0.64	1.61 ± 0.14

**Table 4 microorganisms-11-00948-t004:** Relative fold change in *mecA* gene expression levels of test bacteria **14** and **22**, which developed resistance to oxacillin after 30 days of exposure to aluminum chlorohydrate, and the control group test bacteria taken on the first day as the starter.

Test Bacteria	*mecA*
**22**	1.49 ± 0.59
**14**	5.32 ± 1.46

**Table 5 microorganisms-11-00948-t005:** Relative fold change in *mecA* gene expression levels of test bacteria **14** and **22**, which developed resistance to oxacillin after 30 days of exposure to aluminum chlorohydrate, and the control group test bacteria, which developed resistance to oxacillin following subculturing only in LB broth without aluminum chlorohydrate for 30 days.

Test Bacteria	*mecA*
**22**	4.43 ± 0.91
**14**	6.60 ± 0.91

**Table 6 microorganisms-11-00948-t006:** MIC values of the test bacteria before and after exposure to the test antibiotics (µg/mL).

Test Bacteria	Test Antibiotic	Pre-Exposure	Post-Exposure (30th Day)
**1**	CIP	0.125	4
**2**	OXA	2	16
**3**	OXA	16	>64
**4**	OXA	16	>64
**5**	OXA	0.25	32
**6**	OXA	8	32
**7**	OXA	0.25	16
**8**	OXA	0.25	1
**11**	OXA	1	16
	CIP	0.25	1
**12**	OXA	8	32
**13**	CIP	0.25	8
**14**	OXA	64	>64
**16**	CIP	0.125	0.5
**17**	OXA	0.5	16
**18**	OXA	0.25	64
	CIP	0.25	8
**19**	OXA	0.25	1
**20**	OXA	0.25	1
	CIP	<0.03125	0.125
**21**	OXA	0.25	64
	CIP	0.125	4
**22**	OXA	2	4
	CIP	0.25	4
**23**	OXA	8	64
**24**	OXA	0.125	0.5
	CIP	0.125	0.5

CIP: ciprofloxacin; OXA: oxacillin. **1**–**22**: susceptible isolates; **23**: resistant isolate; **24**: *S. epidermidis* ATCC 12228.

## Data Availability

The data used to support the findings of this study are available from the corresponding author upon request.

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
