# Peer review of "The Effects of Antiperspirant Aluminum Chlorohydrate on the Development of Antibiotic Resistance in Staphylococcus epidermidis"

_microorganisms, 2023, doi:10.3390/microorganisms11040948_

Round 1
Reviewer 1 Report
This study investigated the effects of antiperspirant aluminum chlorohydrate on the development of antibiotic resistance in commensal Staphylococcus epidermidis isolates. It further helps to determine the expression levels of some antibiotic resistance genes, and the minimum inhibitory concentration (MIC) values of the bacteria after the exposure. This study provides new knowledge on the effects of aluminum chlorohydrate as an antiperspirant on the development of antibiotic resistance. It also provides data on the expression levels of some antibiotic resistance genes and the MIC values of bacteria before and after exposure. The results of this study may not be generalizable to other bacteria, as it only focused on Staphylococcus epidermidis. In addition, this study did not determine the mechanism of the observed resistance.
Section 3.1 deserves to be expanded. It is too brief.
Line 291-293: The scientific tone should be adapted here.
What is the context of this ‘These drugs include antidepressants.’? It does not fit here as the work is on antiperspirant/cosmetic. Is this first in history report for antibiotic resistance? Please edit.
Line 323-324 should be expanded. Improve your discussion wrt your topic. This whole section should be revamped.
Last paragraph should go under heading of conclusion.
Author Response
Dear Reviewer,
Thank you for your valuable feedback on our article entitled " The Effects of Antiperspirant Aluminum Chlorohydrate on the Development of Antibiotic Resistance in Staphylococcus epidermidis.” We appreciate your comments and suggestions. Our responses for your comments are below:
- We agree that the study's scope is limited to the effects of antiperspirant aluminum chlorohydrate on commensal Staphylococcus epidermidis isolates, and we have made it clear in the manuscript. We have emphasized the name of the test bacterium in the title, purpose, and throughout the article to clearly state the scope and limitations of our study. We agree with you that the results may not be generalizable to other bacteria. However, we wanted to highlight the possibility of similar resistance development in other commensal bacteria and emphasized the need for further studies in this area.
- Regarding the mechanism of observed resistance, we acknowledge that this was not investigated in the study. However, we believe that the data on the expression levels of some antibiotic resistance genes and the minimum inhibitory concentration (MIC) values of the bacteria after exposure provide significant insights into the effects of aluminum chlorohydrate on the development of antibiotic resistance.
- Section 3.1 is expanded by including the statement "All test bacteria were found to have gyrA, gyrB, parC,parE, norA, and norB/C However, the mecA gene was only detected in test bacteria 1, 12, 13, 14, 15, 16, 22, and 23."
- Line 291-293 is changed as “Microorganisms possess remarkable adaptability to cope with changes in their ecological surroundings. In order to sustain their survival and propagation under these new conditions, they employ a variety of resistance mechanisms.”
- For many years, antibiotic resistance was thought to be caused only by the unnecessary and incorrect use of antibiotics. However, recent studies have shown that non-antibiotic drug classes may also play a role in the development of resistance. Antidepressants are among these drug classes, which is why we emphasized them. Our study is the first in the literature to demonstrate the contribution of aluminum chlorohydrate, used as an antiperspirant, to resistance. Therefore, we discussed the results of our study by giving examples of studies related to antidepressants and non-antibiotic chemicals.
- Line 323-324 is expanded by including the statement “In addition to nonantibiotic drugs, the possibility that daily cosmetics may play a role in developing antibiotic resistance should also be considered. Many cosmetic products contain preservatives and other chemicals designed to prevent microbial growth and prolong the product's shelf life. Therefore, it is important to consider the potential role of cosmetic products in developing antibiotic resistance and further investigate their impact on microbial communities”.
- “Conclusion” title is added to the last paragraph.
Reviewer 2 Report
This was a interesting read and not much research has been done so I would like to thank the authors. I do not have many comments except for that it would be nice to provide citations to the primers listed in Supplementary file. Did the authors design these primers or were they adopted from a previous studies? Do the authors have plans to sequence the genome to study the genetic basis of pathogenecity and virulence of these isolates?
Author Response
Dear Reviewer,
Thank you for taking the time to read our article and provide us with your valuable feedback. We appreciate your positive feedback and are glad that you found our work interesting.
We apologize for the oversight in not including citations to the primers listed in the Supplementary file. We edited the primer citations as references in the table.
Regarding your question about sequencing the genome to study the genetic basis of pathogenicity and virulence of these isolates, we agree that this would be a valuable next step in our research. However, genome sequencing is an expensive process. We have been looking for further support to do this.
Reviewer 3 Report
The concept behind the article “The Effects of Antiperspirant Aluminum Chlorohydrate on the Development of Antibiotic Resistance in Staphylococcus epidermidis” is promising, but the article requires further refinement. The article aims to demonstrate the acquisition of antibiotic resistance in the presence of aluminum chloride, which is commonly used as an antiperspirant in deodorants. The results have the potential to be highly significant for the scientific community, as well as for businesses. I observed that the article lacks proper statistical analysis. Conducting at least two independent experiments is crucial to ensure the findings' validity. By addressing these concerns, the article can become a more impactful and robust piece of research.
Author Response
Dear Reviewer,
Thank you for taking the time to review our article titled "The Effects of Antiperspirant Aluminum Chlorohydrate on the Development of Antibiotic Resistance in Staphylococcus epidermidis.” and for providing us with your valuable feedback.
As you mentioned in your comment,the lack of proper statistical analysis was completed. The One-way ANOVA test was preferred to evaluate the statistical significance between groups and Tukey post-hoc test was also performed to make pairwise comparisons (Figures 3 and 4). Finally, in Figures 5 and 6, paired T-test was performed. Please see the replaced figures and statement in figure and table captions.
Round 2
Reviewer 3 Report
The manuscript has been improved and can be published in this form.